Influence of environmental factors on the spread of COVID-19 in Saudi Arabia

Asdaq Syed Mohammed Basheeruddin sasdaq@gmail.com 1
Rabbani Syed Imam 2
Alamri Abdulhakeem S. 3 4
Alsanie Wala F. 3 4
Alhomrani Majid 3 4
Al-Yamani Mohammad J. 1
1 Department of Pharmacy Practice, College of Pharmacy, AlMaarefa University , Dariyah , Riyadh , Saudi Arabia
2 Department of Pharmacology and Toxicology, College of Pharmacy, Qassim University , Buraydah , Kingdom of Saudi Arabia
3 Centre of Biomedical Sciences Research (CBSR), Deanship of Scientific Research, Taif University, Saudi Arabia , Taif , Saudi Arabia
4 Department of Clinical Laboratory Sciences, The Faculty of Applied Medical Sciences, Taif University, Taif, Saudi Arabia, Taif University , Taif , Saudi Arabia
Palazón-Bru Antonio
Electronic publication date: 2022 Jan 6
Publication date: 2022
Volume: 10
Electronic Location ID: e12732
Received 2021 Oct 20; Accepted 2021 Dec 12
Copyright: ©2022 Asdaq et al.
Copyright year: 2022
Copyright holder: Asdaq et al.
License: This is an open access article distributed under the terms of the Creative Commons Attribution License, which permits unrestricted use, distribution, reproduction and adaptation in any medium and for any purpose provided that it is properly attributed. For attribution, the original author(s), title, publication source (PeerJ) and either DOI or URL of the article must be cited.
License URL: https://creativecommons.org/licenses/by/4.0/

Keywords: Climatic zones, COVID-19, Environmental factors, Saudi Arabia, Temperature, Humidity, Spread

Funding: Taif University no. TURSP 2020/288 Abdulhakeem S. Alamri was supported by Taif University no. TURSP (2020/288). The funders had no role in study design, data collection and analysis, decision to publish, or preparation of the manuscript.

==============================
Background

Coronavirus disease 2019 (COVID-19) has affected millions of people worldwide. The infection is mostly spread through the inhalation of infected droplets. Saudi Arabia is a vast country having different climatic conditions.

Methods

The study evaluated the influence of environmental factors on the spread of COVID-19. Six zones (A to F) were classified depending on the climatic conditions. The study was conducted by retrospective analysis of COVID-19 records from the ministry of health between the months of September 2020 and August 2021. The environmental data such as average temperature (°C), humidity (%), wind speed (m/s) and sun exposure (kwh/m2) were retrieved from official sites. The data was analyzed to determine the effect of these factors on the spread of COVID-19. SPSS IBM 25 software was used to conduct the analysis and p < 0.05 was considered to indicate the significance of the results.

Results

According to the findings, the rate of infection was greater between April and July 2021. Six climatic zones experienced high temperatures, little humidity, consistent wind flow, and intense sun exposure throughout this time. The correlation study revealed a significant (p < 0.05) relationship between the environmental factors and the spread of COVID-19. The data suggested that during summer condition when the weather is hot, less humid, and steady wind flow with lots of sun exposure, the COVID-19 infection rate got augmented in Saudi Arabia. Poor ventilation and closed-door habitats in an air-conditioned atmosphere during this period could have played a role in human transmission. More research on air quality, population mobility and diseased condition is essential, so that precise proactive measures can be designed to limit the spread of infection in specific climatic seasons.

Introduction

Millions of people worldwide were affected by Coronavirus Disease 2019 (COVID-19), which was caused by the severe acute respiratory syndrome coronavirus-2 (SARS-CoV-2). Almost every nation has suffered both economic and human loss due to the spread of the virus. The World Health Organization declared the infection pandemic and recommended several measures to prevent as well as treat the infection (Rothan & Byrareddy, 2020). The infection normally presents with symptoms such as fever, cough, and sore throat, and can cause serious complications in patients suffering from respiratory, cardiovascular, and immunity-related co-morbidities. To date, research at a brisk pace is being conducted to find safe and efficacious preventive and therapeutic interventions for COVID-19 (Zhou et al., 2020).

The major route for person-to-person transmission of COVID-19 is through inhalation of infected air. Infected people’s coughs, sneezes, and speech spread the virus and are thought to be the primary causes of infection spread in the community (Bai et al., 2020). The first case of COVID-19 was reported in the Wuhan city of China in December 2019. The virus rapidly spread to different countries and the first case in Saudi Arabia was reported in March 2020 (Torales et al., 2020). Several factors have been reported for the spread of COVID-19 in the population. The most important among them is the close contact that can happen at places of work, educational institution, markets, religious and private gatherings (Askitas, Tatsiramos & Verheyden, 2021; Coccia, 2021a). The country imposed several precautionary measures according to the guidelines of world health organization to restrict the spread of the infection (Alkahtani et al., 2021). With over 500,000 confirmed cases as of September 2021, the Kingdom of Saudi Arabia is the second most affected country in the Gulf (Al-Tawfiq & Memish, 2020). Currently, the country is witnessing the second wave and has adopted one of the strictest measures in the region to control the infection (Sanders, Monogue & Jodlowski, 2020). Being the largest country in terms of area and population, the reported number of COVID-19 suggests that the government authorities have restricted the spread of infection to a large extent (Al-Turaiki et al., 2021).

The incubation time for presenting the clinical symptoms is reported to be 2–5 days, and practically during this duration, the infected person can transmit the virus to others. The average time between the onset of symptoms and the onset of disease complications is 14 days (Chan et al., 2020). There are several factors that determine the severity of the disease in the patient. Among them are the load of viral exposure, innate immunity of the host, age, and co-morbid conditions of patients (Linton et al., 2020). Earlier studies indicated that environmental factors could also play a role in the dynamics of COVID-19 transmission (Mofijur et al., 2020; Bashir, Bilal & Komal, 2020). Another study suggested that temperatures between 3 °C and 4 °C enhanced the transmission rate in the Chinese population (Meo et al., 2020). Low temperature and high humid conditions were also found to favor the infection rate (Zhu & Xin, 2020; Rosario Denes et al., 2020). Other environmental factors such as wind speed, air quality and total sun exposure has positive effect on the incidences of COVID-19 associated complications (Flaxman et al., 2020; Srivastava, 2021; Diao et al., 2020).

A study conducted in Saudi Arabia suggested that cold weather supports the survival and multiplication of SARS-CoV-2. Their findings indicated that a decrease in temperature, humidity, and wind speed contributed to elevation of COVID-19 cases in Saudi Arabia (Alkhowailed et al., 2020). However, the report also suggests that unstable temperatures could contribute to the outbreak of COVID-19 (Prata, Rodrigues & Bermejo, 2020). Additionally, the data from our previous study revealed that the rate of COVID-19 infection was found to have peaked between the months of May and August in Saudi Arabia, when the temperature during this period normally remains very high (Alharbi et al., 2021). Studies conducted to determine the survival of SARS-CoV-2 observed that even at a high temperature (60 °C), the virus can stay alive for 20 min (Van Doremalen et al., 2020). A detailed analysis is essential to determine the influence of environmental factors on the outbreak of COVID-19 in the population.

Saudi Arabia is a vast country spread between the Red Sea and the Persian Gulf. It consists of deserts, mountains, valleys, forests, and the ocean. The weather and climatic conditions in different regions of Saudi Arabia varies extremely. In some places the weather is cold and in some it is extremely hot (John & Shaiba, 2021). The intense variation in the climatic condition is reported to be due to uneven land elevation above the sea level (Shaukat et al., 2020). The country has thirteen provinces comprised of major cities, towns, and small villages, and, accordingly, the population is distributed either densely or sparsely in these places (Al-Bouwarthan et al., 2019). The region experiences frequent dust storms. The extreme weather conditions with dust storms have been reported to contribute as well complicate the respiratory diseases (Coccia, 2021b). These adverse climatic conditions in the previous studies have been reported to be the risk factors for the COVID-19 complications (Ben Maatoug, Triki & Fazel, 2021). Besides, a significant proportion of population is reported to suffer from the metabolic diseases such as obesity, diabetes mellitus and hypertension. These diseased conditions are also suggested to complicate the COVID-19 (Rahimi et al., 2021).

Analyzing the role played by external factors such as weather conditions in the spread of COVID-19 could assist in designing proactive measures for effective control of infection. Hence, this study was planned to investigate the influence of these factors on the infection rate in different climatic zones of Saudi Arabia.

Methods

Research setting, sample, and data

Saudi Arabia is a vast country having distinct climatic conditions in different regions and is mostly due to variation in land elevation. In some places, it is at 0 m above sea level and in some places as high as 3,000 m. The variation in the elevation of land above sea level has contributed in extreme climatic conditions (John & Shaiba, 2021). After several attempts, a scientific method was adopted to classify the different regions of the country into zones (Fig. 1). These zones were classified depending on the common climatic conditions that prevail throughout the year such as temperature, wind speed, humidity and sun exposure (Alrasheda & Asif, 2015). This scientific method of classification described in the literature was adopted to categorize the regions into six climatic zones such as:

Figure 1 Climatic zones of Saudi Arabia.

Zone-A (Hot-dry with maritime):  Regions included are Dammam, Dahran, Khobar and Hofuf.

Zone-B (Cold-dry with desert): Regions included are Tabuk, Jowf, Arar and Hail.

Zone-C (Hot-dry with desert):  Regions included are Riyadh, Buraidah, Madinah and parts of Makkah.

Zone-D (Hot-dry with maritime desert): Regions included are Jeddah, Jizan, Yanbu and parts of Makkah.

Zone-E (Subtropical with Mediterranean and mountains): Regions included are Taif, Abha, Najran and Khamis mushait

Zone-F (Hot-dry desert with empty quarter): Regions included are Abqaiq, Qatif, parts of Ahsa and parts of Hafar Albatein.

 The rate of infection reported in the official COVID-19 dashboard was collected. This website is linked to the WHO COVID-19 homepage and reports confirmed COVID-19 cases (both symptomatic and asymptomatic). Detailed information about the number of positive cases detected in different parts of the country was retrieved from the official website of the Saudi Arabian Ministry of Health (https://Covid19.cdc.gov.sa/daily-updates/). The number of confirmed cases reported daily for each zone was recorded in an excel sheet and then segregated into respective months. A one-year record of the total number of COVID-19 cases detected in different climatic zones of the country was recorded between September 2020 and August 2021. The data on the weather conditions in different regions of the country was collected from the official website ‘National center for meteorology, KSA’ (https://www.my.gov.sa/wps/portal). The information such as minimum, maximum, and average recorded temperature (°C), humidity (%), wind speed (m/s) and total sun exposure (kwh/m2) was recorded (Kingdom of Saudi Arabia, 2021).

Measure of variables

The environmental parameters were selected as per the literature by Coccia, (2020a). The data retrieved from the previously mentioned sources were analyzed and represented as:

• Normalized infection rate –the total number of COVID-19 infection recorded every month in different climatic zones is normalized (Infection rate = Number of COVID-19 infection / Total population X 10,000) (Yuan et al., 2020) and represented as Fig. 2.

• Mean temperature (°C) –the average temperature measured every month of the climatic zones is represented as Fig. 3.

• Average humidity (%) –the mean humidity measured every month in different regions of the climatic zones is represented as Fig. 4.

• Average wind speed (m/s) –the mean value of wind speed recorded every month in different climatic zones is represented as Fig. 5.

• Total sun exposure (kwh/m2) –the average sun exposure recorded from different regions of the climatic zones in every month is represented as Fig. 6.

Data analysis procedure

The analysis of the data was done as per the methods described by Yuan et al. (2020), Coccia (2021c) and Sarkodie & Owusu (2020). The total number of COVID-19 cases reported in the country was distributed among six climatic zones. The mean weather conditions in different regions of the country were calculated and recorded under respective zones. The COVID-19 and environmental data was represented for each month. The influence of climatic factors such as temperature, humidity, wind speed and total sun exposure on the spread of COVID-19 was analyzed using SPSS-IBM 25 software. Bivariate regression analysis was done between the numbers of COVID-19 cases recorded with the environmental parameters (Fig. 7). Differences between the variables were analyzed by using the Chi-square test. Additionally, Spearman’s Rho correlation analysis was performed to test the association between the environmental parameters on COVID-19 (Table 1) (Alkhowailed et al., 2020). The confidence interval was calculated to determine the precision of estimation. P < 0.05 was used to indicate the significance of results.

Figure 2 Normalized COVID-19 infection rate in different climatic zones.

Figure 3 Recorded mean temperature in different climatic zones.

Results

• In the present study, the normalized rate of COVID-19 infection was recorded from September 2020 to August 2021. The data indicated that from September 2020, the rate of infection progressively declined in all the climatic zones. Zone-C (cold dry with desert) was observed to have the maximum infection rate compared to other zones. The decline continued until December 2020 and January 2021, when the infection rate was found to be at its lowest. From February 2021, the infection rate progressively increased in all zones. Upon comparison, Zone-C (cold dry with desert) recorded the rate of infection, followed by Zone-D (hot-dry with maritime desert) and Zone-A (hot-dry maritime). The peak infection rate recorded for each zone is as follows; Zone-A (hot-dry with maritime) = 10.72 (July 2021), Zone-B (cold-dry with desert) = 8.75 (May 2021), Zone-C (hot dry with desert) = 14.19 (May 2021), Zone-D (hot dry with maritime desert) = 13.28 (May 2021), Zone-E (subtropical with Mediterranean and mountains) = 9.06 (June, 2021) and Zone-F (hot dry desert with empty quarter) = 9.75 (June, 2021). The decline from the peak rate was uniform for all zones except Zone B (cold-dry with desert) and Zone-C (hot dry with desert), where a fluctuation in the infection rate was observed in June 2021, and July 2021. Further, all the climatic zones showed a lower infection rate in the month of August 2021 (Fig. 2).

Figure 4 Humidity level recorded in different climatic zones.

Figure 5 Recorded average wind speed in different climatic zones.

Figure 6 Sun exposure (kwh/m2) measured in different climatic zones.

Figure 7 Bivariate analysis between environmental factors and COVID-19 spread.

• The data on the mean recorded temperature in different climatic zones in a year is represented in Fig. 3. In the month of September 2020, mild to moderate temperature was recorded in all the zones, and the highest and lowest temperatures were found to be in Zone-D (hot dry with maritime desert) (33.7 °C) and Zone-E (subtropical with Mediterranean and mountains) (27.9 °C), respectively. The temperature in the six zones showed progressive reduction reaching lowest in the month of January 2021. The lowest recorded temperature in the month of January 2020 for each zone is as follows; Zone-A (hot dry with maritime) = 14.8 °C, Zone-B (cold dry with desert) = 11.2 °C, Zone-C (hot dry with desert) = 12.6 °C, Zone-D (hot dry with maritime desert) = 26.5 °C, Zone-E (subtropical with Mediterranean and mountains) = 15.5 °C and Zone-F (hot dry desert with empty quarter) = 16.3 °C. From February 2021 the temperature in all the zones increased progressively and reached the peak in the month of August, 2021. The highest recorded temperature in the month of August 2021 is Zone-A (hot dry with maritime) = 36.1 °C, Zone-B (cold dry with desert) = 34.6 °C, Zone-C (hot dry with desert) = 34.9 °C, Zone-D (hot dry with maritime desert) = 35.6 °C, Zone-E (subtropical with Mediterranean and mountains) = 29.6 °C and Zone-F (hot dry desert with empty quarter) = 39.5 °C.

• The record of humidity in different climatic zones of Saudi Arabia indicates that in the month of September 2020 the highest percentage of humidity was observed in the Zone-D (hot dry with maritime desert), while other zones recorded below 40%. The level of humidity progressively increased from September 2020 in all the zones, except in Zone-D (hot dry with maritime desert) where it showed slow decline. The highest level of humidity for Zone-A (hot dry with maritime) was recorded in January, 2021 (59%), Zone-B (cold dry with desert) in January, 2021 (57%), Zone-C (hot dry with desert) in December, 2020 (51%), Zone-D (hot dry with maritime desert) in September, 2021 (67%), Zone-E (subtropical with Mediterranean and mountains) in December, 2020 and January, 2021 (60%) and Zone-F (hot dry desert with empty quarter) in January, 2021 (54%). After reaching peak in January 2021, the percentage of humidity started to decline progressively in all zones. The lowest humidity level was recorded for Zone-C (hot dry with desert) (12%) in January 2021. The record of Zone-D (hot dry with maritime desert) indicated less fluctuation and remained steady in all months (Fig. 4).

• The data for wind speed in different climatic regions is represented in Fig. 5. In the month of September 2020, the highest wind speed was recorded for Zone-A (hot dry with maritime) (3.7 m/s). The wind speed for other zones ranged between 1.7 to 2.3 m/s in the month of September 2020. The wind speed for Zone-A (hot dry with maritime) showed a higher level of fluctuation and the peak for this zone was found in April 2021 (5.8 m/s). The zone-B (cold dry with desert) (5.2 m/s) and Zone-C (hot dry with desert) (2.8 m/s) recorded the highest wind speed in the month of June 2021. Zone-D (hot dry with maritime desert) experienced the highest wind speed in the month of February 2021 (2.6 m/s) and March 2021 (2.5 m/s). Zone-E (subtropical with Mediterranean and mountains) (2.8 m/s) and Zone-F (hot dry desert with empty quarter) (3.1 m/s) recorded their peak wind speed in the month of January 2021.

• The total sun exposure for the different climatic zones can be found in Fig. 6. According to the data available, Zone-B (cold dry with desert) recorded the highest sun exposure followed by Zone-D (hot dry with maritime desert). The least sun exposure was found to be in Zone-E (subtropical with Mediterranean and mountains). The quantum of sun exposure in different zones of climate are Zone-A (hot dry with maritime) (2,000–2,050 kwh/m2), Zone-B (cold dry with desert) (2,300–2,400 kwh/m2), Zone-C (hot dry with desert) (2,100–2,150 kwh/m2), Zone-D (hot dry with maritime desert) (2,150–2,200 kwh/m2), Zone-E (subtropical with Mediterranean and mountains) (1,900–1,950 kwh/m2) and Zone-F (hot dry with empty quarter) (2,050–2,150 kwh/m2).

• The bivariate analysis indicated a significant (p < 0.05) association between the total number of COVID-19 cases in the month of April 2021 and the environmental variables such as temperature and humidity. In the month of May 2021, the correlation was found to be significant (p < 0.05) between COVID-19—temperature and wind speed was found. The analysis for COVID-19 data in the months of June 2021 and July 2021 suggested a significant association (p < 0.05) between all the environmental factors such as temperature, humidity, wind speed and sun exposure on COVID-19 spread (Fig. 7).

Table 1 Summary of correlation analysis between COVID-19 and environmental parameters.

Geographical distribution	Temperature
(Rho value)	Humidity
(Rho value)	Wind speed
(Rho value)	Sun exposure
(Rho value)	
Zone A	−0.136	0.388**	−0.92	−0.201	
Zone B	0.052	−0.132	−0.234	−0.170*	
Zone C	−0.082	−0.267	0.276*	−0.046	
Zone D	−0.159	0.044*	−0.071	−0.336	
Zone E	0.118*	0.164	0.162	−0.172*	
Zone F	−0.188*	−0.074	−0.106	−0.149	
Notes.

Statistics: Spearman’s correlation test.

* p < 0.05.

** p < 0.01.

• Table 1 indicates the correlation between environmental factors and COVID-19 spread in different climatic zones of Saudi Arabia. The analysis of the data between temperature and COVID-19 infection rate indicated a significant (p < 0.05) and positive correlation (Rho = 0.118) for Zone-E (subtropical with Mediterranean and mountains) and, negative and significant correlation (Rho = −0.188, p < 0.05) for Zone-F (hot dry desert with empty quarter). Other zones although suggested both positive and negative correlations but were found to be non-significant. The data for humidity parameter suggested a significant and positive correlation for Zone-A (hot dry with maritime) (Rho = 0.388, p < 0.01) and Zone-D (hot dry with maritime desert) (Rho = 0.044, p < 0.05). However, data of other zones did not suggested similarity in correlation. The wind speed data analysis suggested a positive and significant (Rho = 0.276, p < 0.05) correlation for only Zone-C (hot dry with desert), while other zones were found to be insignificantly correlated. Analysis of the data recorded for total sun exposure in different climatic zones revealed a negative and significant association for Zone-B (cold dry with desert) (Rho = −0.170, p < 0.05) and Zone-E (subtropical with Mediterranean and mountains) (Rho = −0.172, p < 0.05), whereas other climatic zones indicated non-significant correlation.

Discussion

Saudi Arabia is a large country and has an area of 2.3 million square kilometers. The climatic condition of the country varies from region to region and is mainly due to land elevations that range from 0 to 3000 m above the average sea level. According to climatic conditions, the regions of the country are scientifically classified into six zones (Alrasheda & Asif, 2015). The weather varies greatly in these zones, from hot to cold, humid to dry, forest to desert, green lands to empty valleys. Most places receive infrequent rainfall and there is no permanent river (Fig. 1). Water requirements are met through desalination of sea water and, to some extent, underground water (Tarawneh & Chowdhury, 2018).

The present study evaluated the influence of environmental parameters on the spread of COVID-19. The normalized infection rate in Zone-A (hot dry with maritime) started to show a progressive decline from the month of September 2020 and reached its lowest number in December 2020. During this month, the temperature showed a decline, the humidity increased and wind speed slightly fluctuated (Figs. 2 to 5). And from January 2021, the number of COVID-19 cases increased every month and reached its maximum in May 2021. The weather conditions during this period showed an increase in temperature, a decrease in humidity, and an increase in wind speed (Figs. 2 to 5). Zone-A (hot dry with maritime) includes the regions comprised of Dammam, Dahran, Khobar and Hofuf (Almazroui et al., 2012). These places are in hot-dry maritime areas with 2000-2020 kwh/m2 annual sun exposure (Fig. 6) and have mostly oil and gas refinery occupations. The correlation analysis indicated that the level of humidity (Rho = 0.388, p < 0.01) positively influenced the spread of COVID-19 (Table 1). These findings are in accordance with the previous research that suggested a humid atmosphere supports SARS-CoV-2 survival (Alkhowailed et al., 2020; Zhu & Xin, 2020; Rosario Denes et al., 2020). Studies suggest that younger generations have higher chances of infection due to their negligence attitude and reluctance to follow precautionary measures (Zhou et al., 2020).

Data analysis for the climatic Zone-B (cold dry with desert) indicated the lowest COVID-19 infection rate in December 2020 and the corresponding temperature recorded was found to be the lowest (12.1 °C), humidity increased to 53%, and wind speed was steady. The highest infection was observed in July 2021, and the temperature in this month increased to 33.4 °C, with a reduced level of humidity (15%) and moderate wind speed (2.6 m/s) (Figs. 2 to 5). The regions covered under this climatic zone are Tabuk, Jawf, Arar and Hail. These places have cold, dry weather and are mainly covered with deserts (Almazroui et al., 2012). The correlation analysis indicated a negative and significant (Rho = −0.170, p < 0.05) association between sun exposure and COVID-19 spread (Table 1). This zone was found to receive the maximum amount of total sun exposure (2300–2400 kwh/m2) (Fig. 6). Being in a region of extreme cold and hot weather, the reduced exposure to sunlight might have contributed to COVID-19 (Ratnesar-Shumate et al., 2020). Previous studies have indicated that exposure to sun light is essential for the synthesis of vitamin D and for immune activity (Grant et al., 2020). Inadequate levels of vitamin D were observed in hospitalized patients due to COVID-19 (Radujkovic et al., 2020).

In Zone-C (hot dry with desert), the lowest COVID-19 infection rate was recorded in December 2020, and the temperature was found to be reduced to 15.4 °C, with increased humidity (51%) and steady wind speed (2.1 m/s). The highest recorded infection was in June 2021. The temperature in this month increased to 34.7 °C, with reduced humidity (12%) and slightly enhanced wind speed (2.6 m/s) (Figs. 2 to 5). The zone covers the central parts of the country, comprising the hot-dry deserts of Riyadh, Buraidah, Madinah, and parts of Makkah (Almazroui et al., 2012). These regions receive total sun exposure of 2,100–2,150 kwh/m2 (Fig. 6) and rainfall rarely occurs in a year. The correlation analysis indicated a positive and significant association of wind speed (Rho = 0.276, p < 0.05) on COVID-19 (Table 1). As the meteorological data indicated a steady wind flow, the infection might have remained in the atmosphere for a longer duration and contributed to a higher rate of transmission due to the high density of the population in this region. Studies in the past have also suggested low wind flow especially in the densely populated regions enhanced the infection rate (Coccia, 2020b). In addition, high humid conditions and dust storms frequently seen in these regions can augment the pathogenesis of COVID-19 (Coccia, 2021d). Further, dense human population itself can trigger higher transmission of COVID-19 (Rostami et al., 2021; Xu et al., 2020). Polluted air with presence of several toxic chemicals can affect the health of the population and can increase the chances of COVID-19 infection (Sarkodie & Owusu, 2020). Together, it can be suggested that higher population density and mobility with poor air-quality of the region might have influenced the COVID-19 infection rate in this zone (Shen et al., 2021). Similar studies in other parts of world also suggested that the population density, pollution and poor air-quality have enhanced the COVID-19 infection rate (Xu et al., 2020; Yuan et al., 2020).

The climatic Zone-D (hot dry with maritime desert) recorded the highest and lowest COVID-19 infection rate in the month of January 2021 and May 2021, respectively. The corresponding temperature was found to be slightly reduced (26.5 °C) and increased (32.1 °C), while humidity decreased (60%) and then became constant (56%) with steady wind speed during these periods (Figs. 2 to 5). This climatic zone is known for its hot-dry weather with maritime influences and includes cities located on the coast of the Red Sea such as Jeddah, Jizan, Yanbu, and some parts of the Makkah region (Almazroui et al., 2012). The area receives an annual sun exposure of 2,150–2,200 kwh/m2 (Fig. 6). The association between the environmental factors indicated a positive and significant (Rho = 0.044, p < 0.05) correlation between humidity and COVID-19 (Table 1). As the humidity level in this zone remains at a constant level for most of the year, it could have provided a suitable environment for the survival and pathogenesis of COVID-19 (Alkhowailed et al., 2020; Islam et al., 2021).

Analysis of data for Zone-E (subtropical with Mediterranean and mountains) suggested that the lowest recorded COVID-19 infection rate was in December 2021, and the corresponding weather conditions were a drop in temperature (16.5 °C), increased humidity (61%) and steady wind flow (2 m/s). The highest rate of infection was observed in July 2021, with an increased temperature (29.1 °C), decreased humidity (25%) and enhanced wind speed (2.8 m/s) (Figs. 2 to 5). The zone is comprised of high-altitude places with mountains, pleasant weather, water bodies, and greenery. The regions covered are Taif, Abha, Najran, and Khamis Mushait, which are popular tourist spots (Almazroui et al., 2012). The sun exposure was found to be lowest in this zone (1,900–1,950 kwh/m2) (Fig. 6). The correlation analysis suggested a positive and significant association of temperature (Rho = 0.118, p < 0.05), and a negative and significant association of sun exposure (Rho = −0.172, p < 0.05) with COVID-19 spread (Table 1). Earlier studies have suggested that temperature and exposure to sun rays could contribute to the reduction of COVID-19 pathogenesis (Alkhowailed et al., 2020; Grant et al., 2020). The weather conditions of the region are pleasant with low levels of sun exposure, which might have provided the optimal environment for the multiplication of COVID-19 (Almazroui et al., 2012). Being a tourist place, the region could have enhanced the transmission of the infection from one part of the country to another (Askitas, Tatsiramos & Verheyden, 2021).

The data obtained from Zone-F (hot dry desert with empty quarter) indicated the lowest and highest COVID-19 infection rate in January 2021 and June 2021, respectively. Corresponding weather data was temperature decreased in December 2020 (16.3 °C) and increased in June, 2021 (34.8 °C), humidity increased (54%) and then decreased (38%) and wind speed was 2.1 m/s in December, 2020 and 3.1 m/s in January, 2021 (Figs. 2 to 5). This climatic zone is known as the “empty zone” as the human habitat is sparingly distributed in the vast areas of deserts. The regions covered under this area are Abqaiq, Qatif, parts of Ahsa and parts of Hafar Albatein (Almazroui et al., 2012). The total sun exposure received here was 2050–2150 kwh/m2 (Fig. 6). The correlation analysis suggested a negative and significant association between temperature (Rho = −0.188, p < 0.05) for the COVID-19 (Table 1). The region is comprised of deserts with extreme temperatures that might have contributed to most of the indoor activities with a closed atmosphere, leading to the spread of COVID-19 (Alkhowailed et al., 2020; Grant et al., 2020).

Overall, the data from the present study indicated that the rate of infection increased in all the climatic zones when the weather becomes hot, less humid, and steady wind flow. Temperature is one of the important characteristics that is reported to directly influence the immune activity of the host system (Alkhowailed et al., 2020). Extreme temperature conditions (both cold and hot) are known to support microbial infections (Alkhowailed et al., 2020; Ahmadi et al., 2020). However, our data suggested that at lower temperatures, the infection rate declined, and as the temperature increased, the infection rate also increased (Figs. 2 to 3). One reason for this could be the closed-door activities preferred during the summer season in this region.

The major portion of Saudi Arabia is reported to experience a peak temperature between the months of May and August every year (Haque & Rahman, 2020). Further, the lower humidity level experienced during this period (Fig. 4) prevents ventilation and makes it uncomfortable to venture out in scorching heat. During this period, the population prefers to stay and work in a closed-door atmosphere. The use of air-conditioning has also become routine during this season, preventing ventilation, and contributing to a higher transmission rate of COVID-19 (Rule, 2020). Further, wind speed in most of the climatic zone was found to be steady except in coastal zones such as Zone-A (hot dry with maritime) and part of Zone-B (cold dry with desert) (Fig. 5). Slightly more than average wind speed during the months of May–August (Fig. 5) might have augmented the chances of transmission when people in the evening ventured out of their residences.

According to the literature, the amount of sun exposure is directly responsible for the antimicrobial activity. The powerful rays from the sun, such as ultraviolet, can cause damage to microbial cell walls, decreasing the chance of infection (Grant et al., 2020). The data from the study indicated that most of the country received an above-normal exposure to the sun, and this can be seen in most parts of the year (Fig. 6). However, the maximum sun exposure was found to be during the summer season when the days are long and hot. The summer in Saudi Arabia is extreme in most places except in Zone-E (subtropical with Mediterranean and mountains) (Almazroui et al., 2012). Since a major proportion of the population avoids sun exposure during summer, this might have prevented the chances of the virus being killed due to sun light and the preference to stay indoors might have increased the transmission rate in summer (Rule, 2020; Coccia, 2020c). The bivariate analysis also indicated the correlation of environmental factors such as temperature, humidity, wind speed, and sun exposure on the COVID-19 transmission between the months of peak summer such as April 2021, and July 2021 (Fig. 7).

The decline in the COVID-19 infection from the month of August 2021 could be related to several factors. Saudi Arabia is known to have implemented one of the strictest precautionary measures such as wearing of face marks, social distancing, preventing crowding in public places and avoiding social gatherings (Alkahtani et al., 2021). The country is providing free COVID-19 immunization for people above 12 years age. The vaccination program started from the beginning of 2021 and has achieved herd immunity in October 2021 in all regions of climatic zones. The health authorities of the country were reported to follow updated guidelines for preventing and treating the COVID-19 infection (Al-Turaiki et al., 2021). These factors could also be the reason for decline in the infection.

The observations of the present study have provided a preliminary data on the spread of COVID-19 in six climatic zones of Saudi Arabia. Although, biological characteristics of the SARS-CoV-2 does not change between regions but its transmission and infection rate are reported to get influenced by the environmental and population-based factors (Coccia, 2021e). In the absence of substantial evidence to link the role of specific environmental factor, more studies are suggested with an aim to determine the role of air-quality, population mobility and diseased conditions of the population on the COVID-19 spread (Coccia, 2021f).

Conclusions

The current research examined the environmental factors that would have influenced the spread of COVID-19 in Saudi Arabia’s six climatic zones. According to the data, humid conditions in maritime zones, high temperature in hot-dry zones, inadequate sun exposure in cold-dry zones and tourist activities in subtropical zones might have influenced the COVID-19 infection rate. All the zones experienced peak infection rate between the months April 2021 and July 2021 that corresponds to summer condition in the country. Close-door activities in hot-dry places, insufficient ventilation in humid regions, inadequate sun exposure in cold-dry areas and movements of travelers in the subtropical Mediterranean quarters might have augmented the transmission rate of COVID-19 infection. The healthcare authorities and public can have a look on these aspects and can design suitable strategies to avoid close-door activities, improve the ventilation, increase the sun exposure and restrict the vocational events to prevent the next wave of pandemic in their regions of habitats. In the absence of specific link between other factors such as role of air-quality, diseased states and population mobility, more research is suggested in this direction to determine their influence on the rate of COVID-19 infection.

Supplemental Information

Supplemental Information 1 Raw data

Click here for additional data file.

Additional Information and Declarations

Competing Interests

Author Contributions

Data Availability

The authors declare there are no competing interests.

Syed Mohammed Basheeruddin Asdaq conceived and designed the experiments, performed the experiments, authored or reviewed drafts of the paper, and approved the final draft.

Syed Imam Rabbani conceived and designed the experiments, performed the experiments, prepared figures and/or tables, and approved the final draft.

Abdulhakeem S. Alamri conceived and designed the experiments, analyzed the data, prepared figures and/or tables, and approved the final draft.

Wala F. Alsanie and Majid Alhomrani conceived and designed the experiments, analyzed the data, authored or reviewed drafts of the paper, and approved the final draft.

Mohammad J. Al-Yamani analyzed the data, authored or reviewed drafts of the paper, and approved the final draft.

The following information was supplied regarding data availability:

The raw data in the Supplemental File was taken from the Public Health Authority’s Daily Updates: https://covid19.cdc.gov.sa/daily-updates/.

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
