# Peer review of "Influence of environmental factors on the spread of COVID-19 in Saudi Arabia"

_PeerJ, doi:10.7717/peerj.12732_

## Round 0.1 · original submission · Major Revisions

There are major issues which the authors should address in a revised version of the text.
Please, consider only the suggested references with scientific value for your work. If you think that these suggestions are not suitable for the manuscript, you can explain why and I will carefully review that part, in case the reviewer does not agree with omitting the references. Many thanks

Reviewer 1 ·

Basic reporting

Zones have to be clarified.
see additional comments

Experimental design

It can be improved with a clear structure.
Zones have to be clarified.
see additional comments

Validity of the findings

It can be imporved in the visual presentation with figures.
Zones have to be clarified.
see additional comments

Additional comments

Investigating the environmental factors that influenced the spread of Covid-19 in Saudi Arabia

The topics of this paper are interesting, but the structure and content must be revised, and results have to be better explained by authors before to be reconsidered for publication.

Title has to be shorter. COVID-19 and not Covid-19 in all text.

Abstract has to be shorter focusing on results and environmental and social implications. The presentation of all these zones create confusion…try to clarify that these are climate zones and if the COVID-19 is higher in dry or wet zones, hot or cold zones etc.….

Introduction has to better clarify the research questions of this study and provide more theoretical background. Authors have to better describe the different sources of transmission dynamics of COVID-19 (e.g., mobility, environment, pollution, etc.) and risk factors in society, that can accelerate diffusion of this novel coronavirus in this country (See suggested readings that must be all read and used in the text).

Methods of this study is not clear. The section of Materials and methods must be re-structured with following three sections:
• Research setting, sample and data
• Measures of variables
• Data analysis procedure.

Results. Authors have to avoid subheadings that create fragmentation and confusion. If necessary, can use bullet points.
Table 1 can remove the columns with p, because significance is indicated with *, **. Also, lower and higher 95% can be removed because results can be redundant. Table has to be simple.

Figure 1 has to insert a note clarifying the geoclimatic zones. To explain them in the text is not enough for clearness for readers. Total number should be normalized with total swabs or population within zones under study.

Figure 2 specify on y-axis that it is temperature in °C
For all figures, authors have to insert a note clarifying the geoclimatic zones. As said, to explain them in the text is not enough for clearness for readers.

Discussion has to consider suggested papers to reinforce results on wind speed, air pollution and diffusion of COVID-19.

Conclusion is too short and has no sense….it has not to be a summary, but authors have to focus on manifold limitations of this study and provide suggestions of environmental, health, crisis management and social policy. In particular, authors have to suggest best responses of prevention and preparedness of next waves of COVID-19 and similar pandemics.

Overall, then, the paper is interesting. Theoretical framework is weak, and some results create confusion… structure of the paper has to be improved; study design, discussion and presentation of results have to be clarified using suggested comments.

If the paper is improved, by using all comments, maybe it can be considered.

Suggested readings of relevant papers that have to be read and all inserted in the text and references.

Meo, S.A., Abukhalaf, A.A., Alomar, A.A., (...), Al-Khlaiwi, T., Usmani, A.M. 2020 Effect of temperature and humidity on the dynamics of daily new cases and deaths due to COVID-19 outbreak in Gulf countries in Middle East Region, European Review for Medical and Pharmacological Sciences 24(13), pp. 7524-7533

Zhu Y., Xin J. 2020. Association between ambient temperature and COVID-19 infection in 122 cities from China, Science of the Total Environment, https://doi.org/10.1016/j.scitotenv.2020.138201

Srivastava, A. 2021. COVID-19 and air pollution and meteorology-an intricate relationship: A review, Chemosphere, 263,128297

Coccia M. 2021. Preparedness of countries to face covid-19 pandemic crisis: Strategic positioning and underlying structural factors to support strategies of prevention of pandemic threats, Environmental Research, n. 111678, https://doi.org/10.1016/j.envres.2021.111678

Bashir, M.F., Bilal, B.M., Komal, B., 2020. Correlation between environmental pollution indicators and COVID-19 pandemic: A brief study in Californian context. Environ. Res. 109652.https://doi.org/10.1016/j.envres.2020.109652

Xu K, Cui K, Young L-H, Hsieh Y-K, Wang Y-F, Zhang J, et al. Impact of the COVID-19 Event on Air Quality in Central China. Aerosol and Air Quality Research 2020b; 20: 915-929.

Coccia M. 2021. The impact of first and second wave of the COVID-19 pandemic: comparative analysis to support control measures to cope with negative effects of future infectious diseases in society. Environmental Research, vol. 197, June, n. 111099, https://doi.org/10.1016/j.envres.2021.111099

Rosario Denes, K.A., Mutz Yhan, S., Bernardes Patricia, C., Conte-Junior Carlos, A., 2020. Relationship between COVID-19 and weather: case study in a tropical country. Int. J. Hyg Environ. Health 229, 113587.

Coccia M. 2021. Comparative Critical Decisions in Management. In: Farazmand A. (eds), Global Encyclopedia of Public Administration, Public Policy, and Governance. Springer Nature, Cham. https://doi.org/10.1007/978-3-319-31816-5_3969-1

Askitas, N., Tatsiramos, K., Verheyden, B. 2021. Estimating worldwide effects of non-pharmaceutical interventions on COVID-19 incidence and population mobility patterns using a multiple-event study (Open Access)(2021) Scientific Reports, 11 (1), art. no. 1972.

Coccia M. 2020. Factors determining the diffusion of COVID-19 and suggested strategy to prevent future accelerated viral infectivity similar to COVID, Science of the Total Environment, Number: 138474. https://doi.org/10.1016/j.scitotenv.2020.138474

Shen, L., Zhao, T., Wang, H., (...), Zhu, Y., Shu, Z. 2021Importance of meteorology in air pollution events during the city lockdown for COVID-19 in Hubei Province, Central China Science of the Total Environment, 754,142227

Haque, S.E., Rahman, M. 2020. Association between temperature, humidity, and COVID-19 outbreaks in Bangladesh, Environmental Science and Policy, 114, pp. 253-255

Yuan, J., Li, M., Lv, G., Lu, Z.K. 2020. Monitoring transmissibility and mortality of COVID-19 in Europe ((2020) International Journal of Infectious Diseases, 95, pp. 311-315.

Coccia M. 2020. An index to quantify environmental risk of exposure to future epidemics of the COVID-19 and similar viral agents: Theory and Practice. Environmental Research, n. 110155, DOI: 10.1016/j.envres.2020.110155

Sarkodie, S.A., Owusu, P.A. 2020. Impact of meteorological factors on COVID-19 pandemic: Evidence from top 20 countries with confirmed cases. Environmental Research, 191,110101

Coccia M. 2021. The effects of atmospheric stability with low wind speed and of air pollution on the accelerated transmission dynamics of COVID-19. Journal: International Journal of Environmental Studies, vol. 78, n. 1, pp. 1-27, https://doi.org/10.1080/00207233.2020.1802937

Flaxman, S., Mishra, S., Gandy, A., Unwin, H.J.T., Mellan, T.A., Coupland, H., Whittaker, C., (...), Bhatt, S. 2020. Estimating the effects of non-pharmaceutical interventions on COVID-19 in Europe (Open Access), (2020) Nature, 584 (7820), pp. 257-261.

Diao, Y., Kodera, S., Anzai, D., (...), Rashed, E.A., Hirata, A. 2021. Influence of population density, temperature, and absolute humidity on spread and decay durations of COVID-19: A comparative study of scenarios in China, England, Germany, and Japan, One Health, 12,100203

Coccia M. 2021. How do low wind speeds and high levels of air pollution support the spread of COVID-19? Atmospheric Pollution Research, vol. 12, n.1, pp. 437-445., https://doi.org/10.1016/j.apr.2020.10.002.

Rahimi, N.R., Fouladi-Fard, R., Aali, R., (...), Conti Gea, O., Fiore, M. 2021 Bidirectional association between COVID-19 and the environment: A systematic review Environmental Research, 194,110692

Islam, N., Bukhari, Q., Jameel, Y., (...), Massaro, J.M., D'Agostino, R.B. 2021, COVID-19 and climatic factors: A global analysis. Environmental Research, 193,110355

Coccia M. 2020. How (Un)sustainable Environments are Related to the Diffusion of COVID-19: The Relation between Coronavirus Disease 2019, Air Pollution, Wind Resource and Energy. Sustainability 2020, 12, 9709; doi:10.3390/su12229709

Alkahtani, T.A., Alakeel, A., Alakeel, R.A., (...), Abdel-Daim, M.M., Jammah, A.A. 2021. The current reproduction number of COVID-19 in Saudi Arabia: is the disease controlled? Environmental Science and Pollution Research28(33), pp. 44812-44817

Coccia M. 2021. The relation between length of lockdown, numbers of infected people and deaths of Covid-19, and economic growth of countries: Lessons learned to cope with future pandemics similar to Covid-19. Science of The Total Environment, n. 145801. https://doi.org/10.1016/j.scitotenv.2021.145801

Al-Turaiki, I., Almutlaq, F., Alrasheed, H., Alballa, N. 2021. Empirical evaluation of alternative time-series models for covid-19 forecasting in Saudi Arabia, International Journal of Environmental Research and Public Health 18(16),8660

John, M., Shaiba, H. 2021 Correlation between weather and COVID-19 cases: An extensive study covering all provinces in Saudi Arabia2021 International Conference of Women in Data Science at Taif University, WiDSTaif 2021, 9430202

Coccia M. 2021. Effects of the spread of COVID-19 on public health of polluted cities: results of the first wave for explaining the dejà vu in the second wave of COVID-19 pandemic and epidemics of future vital agents. Environmental Science and Pollution Research. 28(15), 19147-19154. https://doi.org/10.1007/s11356-020-11662-7

Ben Maatoug, A., Triki, M.B., Fazel, H. 2021How do air pollution and meteorological parameters contribute to the spread of COVID-19 in Saudi Arabia? Environmental Science and Pollution Research, in press

Alsayedahmed, H.H. 2020 COVID-19 Pandemic’s precautionary measures had hit the reset button of the quality of life at different aspects, Journal of Infection in Developing Countries14(8), pp. 812-816

Coccia M. 2021. Pandemic Prevention: Lessons from COVID-19. Encyclopedia of COVID-19, 1, pp. 433–444. https://doi.org/10.3390/encyclopedia1020036

Alkhowailed, M., Shariq, A., Alqossayir, F., (...), Rasheed, Z., Al Abdulmonem, W. 2020. Impact of meteorological parameters on COVID-19 pandemic: A comprehensive study from Saudi Arabia: Impact of weather on COVID-19, Informatics in Medicine Unlocked20,100418

Coccia M. 2021. High health expenditures and low exposure of population to air pollution as critical factors that can reduce fatality rate in COVID-19 pandemic crisis: a global analysis. Environmental Research, vol. 199, Article number 111339, https://doi.org/10.1016/j.envres.2021.111339

Meo, S.A., Abukhalaf, A.A., Alomar, A.A., (...), Usmani, A.M., Akram, J. 2020Climate and COVID-19 pandemic: Effect of heat and humidity on the incidence and mortality in world's top ten hottest and top ten coldest countries, European Review for Medical and Pharmacological Sciences 24(15), pp. 8232-8238

Reviewer 2 ·

Basic reporting

Perhaps the best part of this paper is the introductory review, even if there is a clear mistake in the year of reference of Sanders, put before the outbreak started. Instead the results are not convincing in suppori of the hypothesis of a possible correlation. If a correlation exists it cannot depend on the geographical area, unless this fact finds an explanation.

Experimental design

No comment about the idea of looking for correlations between the diffusion of the pandemic and climatica variable

Validity of the findings

The outbreak in Saudi Arabia is mainly due to two regions, thus the subdivision in six does not pprovide much advantage. The period indicated as one of growing cases (May-August 2021) registered an average of 1100 new cases per day, not much different from the previous fourtyfive days (915) whereas in the two previous months the contaions were significantly less (about 350) and in the following two months even less (about 100). The regions are very similar for what concerns tempperature (with one exception) and similar consoderations are valid for the other variables. There are large differences in thenumber of ontagions between the different regions. This situation makes very doubtful that one can assess the existence of correlations through analysis for different regions. If a correlation exists it should be exhibited everywhere, unless thee are reasons explaining the difference. This seems not to be the case for the difference, with respect to T of the regions E and F. A possible explanation might be found in the fact that both these region have a much smaller number of cases. making their results less significan than those of recions like C. Similar considerations can be made for the pther variables. The indication of the periods when each variabke reached maximum and minimum values seems irrelevamt to deaw conclusions, that instead seem to have been obtained more on the basis of authors expectations than on scientific results.

Additional comments

No further comments

Reviewer 3 ·

Basic reporting

The manuscript subject “Investigating the environmental factors that influenced the spread of Covid-19 in Saudi Arabia” is interesting. However, the manuscript needs improvement. The authors made a considerable effort, but unfortunately, there are severe flaws in the analysis, which affect all results.

• The role of lockdown and other restrictions also affect the results that are not determined in the manuscript, so the reason for the fluctuation in the Covid-19 outbreak is not just the climate. The periods and locations with the same restrictions (necessity of facemask, lockdown period, etc.) can be compared.
• Moreover, there would be no correlations between the daily COVID-19 cases and the weather data of the same day, and the achieved results are wrong. The estimated incubation period of COVID-19 is about 2-14 days. The positive cases mean the positive test of the cases with symptoms such as a fever, cough, or shortness of breath, and the symptoms of COVID-19 occur about 3 to 5 days. Therefore, there is a delay among the weather data and COVID-19 cases. Moreover, there are two other delays, including the laboratory analysis results and the announced time.
• In addition, the start of the vaccination program is not mentioned in the paper. Until 21 August, was the vaccination zero, or was it an equal rate in all compared zones?

Experimental design

there are severe flaws in the analysis

Validity of the findings

there are severe flaws in the analysis

Additional comments

there are severe flaws in the analysis

---

## Round 0.2 · accepted · Accept

After analyzing the authors' response to the reviewers' comments, from my point of view I can see a relevant improvement with scientific validity, therefore I am pleased to accept your manuscript for publication in PeerJ. Congratulations!

Reviewer 1 ·

Basic reporting

It is OK.

Experimental design

It is OK.

Validity of the findings

It is OK.

Additional comments

I have read thoroughly the revised version of paper.

The authors have done considerable additional work, and addressed all concerns and criticisms in the revised manuscript, which I believe has improved substantially in the theoretical framework, study design and discussion of results.

Now, the paper is OK and has a good level to show interesting results to scholars and/or policymakers interested in these topics to solve problems of COVID-19 pandemic crisis.